# Correlation between Endophthalmitis and Stroke Development in Ankylosing Spondylitis Patients: A Population-Based Cohort Study

**DOI:** 10.3390/ijerph192013108

**Published:** 2022-10-12

**Authors:** Yung-En Tsai, Wu-Chien Chien, Yi-Hao Chen, Chi-Hsiang Chung, Jiann-Torng Chen, Ching-Long Chen

**Affiliations:** 1Department of Ophthalmology, Kaohsiung Armed Forces General Hospital, Kaohsiung 802, Taiwan; 2Department of Ophthalmology, Tri-Service General Hospital, National Defense Medical Center, Taipei 114, Taiwan; 3Department of Medical Research, Tri-Service General Hospital, National Defense Medical Center, Taipei 11490, Taiwan; 4School of Public Health, National Defense Medical Center, Taipei 11490, Taiwan; 5Graduate Institute of Life Sciences, National Defense Medical Center, Taipei 11490, Taiwan; 6Taiwanese Injury Prevention and Safety Promotion Association, Taipei 11490, Taiwan

**Keywords:** endophthalmitis, stroke, ankylosing spondylitis, cohort study

## Abstract

Background: This cohort study aimed to research the correlation between endophthalmitis and stroke development in ankylosing spondylitis (AS) patients by reviewing National Health Insurance Research Database (NHIRD) data. Methods: This study obtained data from the NHIRD over a sixteen-year period. The primary outcome was stroke development. We used Fisher’s exact test and Pearson’s chi-squared test to analyze the variables. We investigated the risk factors for disease development using Cox regression analyses. We compared the cumulative incidence of stroke using Kaplan–Meier analysis. Results: The study cohort included 549 patients with AS and endophthalmitis, while the comparison cohort included 2196 patients with AS but without endophthalmitis. The stroke development was increased in the study cohort (adjusted hazard ratio, 1.873; *p* ≤ 0.001). The total stroke development in the study cohort and the comparison cohort was 1724.44 per 100,000 person-years and 1085.11 per 100,000 person-years, respectively (adjusted hazard ratio, 1.873; 95% confidence interval, 1.776–2.022; *p* < 0.001). Our study cohort showed an increased stroke rate. Conclusions: Our studies showed that endophthalmitis increases the risk of stroke in AS patients and endophthalmitis is an independent risk factor for stroke in AS patients. Nonetheless, advanced studies that thoroughly investigate the correlation between endophthalmitis and stroke in AS patients are needed to validate our findings.

## 1. Introduction

Ankylosing spondylitis (AS) is a chronic inflammatory disorder that primarily affects the axial skeleton, such as the spine and sacroiliac joints, of primarily young men. AS is classified as seronegative spondyloarthropathy, which often presents as joint pain with stiffness that improves with activity. Although several extra-articular features are associated with AS, such as uveitis, inflammatory bowel disease, and bone, skin, heart, lung, and kidney involvement, stroke was well documented in recent studies [1,2,3,4]. AS patients are at increased risk of stroke. It is possible that inflammatory processes are involved in AS. Inflammation in plaque formation and atherogenesis, which causes cerebrovascular and cardiovascular events, is recognized [5].

Clinically, stroke is an acute event of focal dysfunction of the brain, spinal cord, or retina lasting longer than 24 h or of any duration if imaging reveals focal hemorrhage or infarction corresponding to the symptoms [6]. Several risk factors have been cited for stroke, including infection and inflammation. Infections caused by bacteria, viruses, fungi, and parasites can cause stroke via endogenous blood invasion or direct exogenous invasion. Moreover, the infection-induced inflammatory response causes a procoagulant state associated with stroke risk [7,8]. Many infectious diseases are associated with stroke, such as infective endocarditis, bacterial meningitis, tuberculous meningitis, neurosyphilis, neuroborreliosis, rickettsial diseases, virus-induced vasculitis or vasculopathy, fungal meningitis, and mucormycosis [8]. However, endophthalmitis with stroke was discussed only in a case report [9].

Endophthalmitis, a panuveitis, is an inflammation affecting the entire eye, including the anterior and posterior segments. Ocular tissue damage can be mediated by inflammatory byproducts or directly by the invading organism itself [10]. AS patients with systemic infection can develop complications, as endophthalmitis has been reported in some case reports [11,12]. Moreover, a previous case report demonstrated the correlation between endophthalmitis and stroke [9]. However, data are scant about the correlation between endophthalmitis and stroke in AS patients.

Thus, our study aimed to research the correlation between endophthalmitis and stroke development in AS patients by reviewing National Health Insurance Research Database (NHIRD) of Taiwanese data.

## 2. Materials and Methods

### 2.1. Research Database

The National Health Insurance (NHI) program, which covers almost 99% of the Taiwanese population (currently approximately 23 million people), was launched in Taiwan in 1995. The NHIRD contains claims data associated with patients enrolled in the NHI program. In the NHIRD, fundamental data, such as sex, age, diagnosis, and comorbidities, can be accessed. These data can be used for statistical research in an electronic format. Therefore, we used this database to research the correlation between endophthalmitis and stroke in AS patients.

### 2.2. Study Participants

In this retrospective cohort study, we identified patients with AS in the NHIRD using the International Classification of Disease, Ninth Revision, Clinical Modification (ICD-9-CM) code (720.0). In this study, AS was an inclusion criterion. The exclusion criteria were diagnosis with AS before the inclusion date, stroke before tracking, lack of tracking, age < 20 years, and unknown sex. We further extracted the ICD-9-CM codes for endophthalmitis (360.0, 360.00–360.04, and 360.1) among the identified AS patients as the study cohort. We enrolled 21,846 patients who conformed to the inclusion criteria, from 1 January 2000 to 31 December 2015, but we excluded 2140 patients in accordance with the exclusion criteria. Our study population included 19,706 patients. Among our study population, the study cohort included 549 patients with endophthalmitis. We collected the comparison cohort with the criteria of the study cohort utilizing a fourfold propensity score, which matched by sex, age, comorbidities, and index date. We included a total of 2196 patients in the comparison cohort. Amid the sixteen-year period, 318 patients had diagnosed stroke, including 89 in the study cohort and 229 in the comparison cohort (Figure 1).

### 2.3. Ethical Considerations

To protect patient privacy, the NHIRD encodes information of personal patients. Thus, written patient consent was not required.

### 2.4. Statistical Analysis

We analyzed the characteristics of AS patients with and without endophthalmitis at baseline versus study end. We compared continuous variables between cohorts using the independent Student’s *t*-test. We calculated the differences in the categorical variables with statistical significance defined as *p* < 0.05 using Fisher’s exact test and Pearson’s chi-squared test. We performed uni- and multivariate Cox regression analyses after adjustment of the variables to estimate the adjusted hazard ratio (aHR) for stroke risk. Considering the risks, we performed Cox regression for the analysis. We also predicted the cumulative incidence of stroke in these two cohorts by Kaplan–Meier analysis. We performed all statistical analyses using SPSS version 22 (SPSS Inc., Chicago, IL, USA).

## 3. Results

The demographic characteristics of the cohort are presented in Table 1. A total of 2745 individuals were enrolled, including 549 (20%) individuals with AS and endophthalmitis and 2196 (80%) with AS but without endophthalmitis. The mean ages of the AS with endophthalmitis and AS without endophthalmitis groups were 37.00 ± 18.65 years and 37.08 ± 19.80 years, respectively. There were no significant intergroup differences in sex, age, asthma, atrial fibrillation (Af), cardiomegaly, chronic obstructive pulmonary disease (COPD), congestive heart failure (CHF), coronary artery disease (CAD), diabetes mellitus (DM), hyperlipidemia, hypertension (HTN), and Charlson comorbidity index revised (CCI_R) scores (Figure 2). At the tracking endpoint, 89 (16.21%) and 229 (10.43%) patients had and did not have endophthalmitis, respectively (*p* < 0.001). At the endpoint, the mean ages were 41.24 ± 19.78 years among the AS patients with endophthalmitis and 41.53 ± 20.02 among the AS patients without endophthalmitis. In AS patients with endophthalmitis, rates of stroke, asthma, Af, COPD, CHF, DM, HTN, CCI_R score, and all-cause mortality were increased. The demographic characteristics of this study at the endpoint are presented in Table 2, while the comorbidities of the study population at the endpoint are presented in Figure 3.

The mean follow-up duration in the AS with endophthalmitis and AS without endophthalmitis groups were 9.42 ± 8.31 years and 9.60 ± 8.42 years, respectively, showing no intergroup difference (Appendix A). The mean timing of developing stroke after enrollment was 4.24 ± 4.65 years in the AS with endophthalmitis group versus 4.65 ± 4.71 years in the AS without endophthalmitis group (*p* < 0.001; Appendix A).

The Kaplan–Meier method was used to calculate the cumulative risk of stroke development (Figure 4A). We also divided stroke into hemorrhagic (Figure 4B) and ischemic (Figure 4C) types. Our study cohort had a greatly increased cumulative risk of stroke, hemorrhagic stroke, and ischemic stroke (log-rank test, *p* < 0.001).

Table 3 presents the risk factors for stroke using Cox regression analysis. The aHR for endophthalmitis, male sex, age 40–59 years, age ≥ 60 years, asthma, Af, COPD, CHF, CAD, DM, hyperlipidemia, HTN, and CCI_R scores were 1.873, 1.602, 1.553, 1.843, 2.076, 1.267, 1.803, 2.286, 3.031, 2.572, 2.410, 2.623, and 1.625, respectively (all *p* < 0.001). Figure 5 shows forest plots of the crude hazard ratio and aHR for stroke factors evaluated in Table 3.

In Table 4, comparing AS patients with and without endophthalmitis in the stratified analysis, the stroke development was 1724.44 per 100,000 person-years in the study cohort and 1085.11 per 100,000 person-years in the comparison cohort (aHR = 1.873; 95% CI, 1.776–2.022; *p* < 0.001). The AS patients with endophthalmitis were at an increased risk of stroke development regardless of stratified variables (sex, age, asthma, Af, cardiomegaly, COPD, CHF, CAD, DM, hyperlipidemia, and HTN).

In Table 5, we divided stroke into hemorrhagic and ischemic stroke and compared AS patients with and without endophthalmitis. The hemorrhagic stroke development was 909.00 per 100,000 person-years in the study cohort and 563.88 per 100,000 person-years in the comparison cohort (aHR = 1.998; 95% CI, 1.895–2.153; *p* < 0.001). The ischemic stroke development was 812.30 per 100,000 person-years in the study cohort and 521.23 per 100,000 person-years in the comparison cohort (aHR = 1.794; 95% CI, 1.692–1.924, *p* < 0.001). Patients with AS and endophthalmitis are increasing risk of developing hemorrhagic or ischemic stroke.

## 4. Discussion

Our study enrolled 549 patients in the study cohort and 2196 patients in the comparison cohort. This revealed that the risk of stroke development was significantly increased in this study versus the comparison cohort. In accordance with the Kaplan–Meier analysis, the cumulative risk of developing stroke, hemorrhagic stroke, or ischemic stroke was increased in AS patients with endophthalmitis. Additionally, the significant risk factors for stroke development in AS patients included the 40–59 years and ≥60 years age groups, asthma, Af, COPD, CHF, CAD, DM, hyperlipidemia, HTN, and higher CCI_R score. In addition, AS patients with endophthalmitis and comorbidities had a higher aHR for developing stroke. Moreover, age 40–59 years was the main risk factor for stroke in AS patients with endophthalmitis. To the best of our knowledge, no previous studies have shown a correlation between endophthalmitis and the stroke development in AS patients.

Current evidence reveals an increased incidence of stroke in AS patients [1,2,3,4]. Han et al. [1] reported that cerebrovascular disease risk was increased in AS patients (relative risk, 1.7; 95% CI, 1.3–2.3). Szabo et al. [2] reported that cerebrovascular disease risk was increased in AS (standardized prevalence ratio, 1.3; 95% CI, 1.2–1.4). Keller et al. [3] reported that stroke risk was increased in AS (hazard ratio, 2.4; 95% CI, 2.0–2.8). Liu et al. [4] reported that stroke risk was greatly increased in AS (relative risk, 1.49; 95% CI, 1.25–1.77). However, two studies revealed no increased incidence of stroke in AS patients [13,14]. AS and endophthalmitis in our study led to a 1.873-fold increased risk of stroke (95% CI, 1.776–2.022) compared to AS patients without endophthalmitis. Nonetheless, compared to other studies, we should carefully interpret our results.

In our study, men with AS were at a higher risk of developing stroke than women with AS. Patients aged 40–59 years and ≥60 years were at a higher risk of stroke development than those aged 20–39 years. Moreover, AS patients with asthma, Af, COPD, CHF, CAD, DM, hyperlipidemia, or HTN were at higher risk of developing stroke (Table 3). Male sex [15], older age [15], asthma [16], Af [17], COPD [18], CHF [19], CAD [20], DM [21], hyperlipidemia [22], and HTN [23] are risk factors for stroke in the general population. Nonetheless, no other study has investigated the risk factors for stroke in patients with AS. Consistent with previous studies [15,16,17,18,19,20,21,22,23], our study showed that male sex, older age, asthma, Af, COPD, CHF, CAD, DM, hyperlipidemia, and HTN were also risk factors for stroke development among AS patients.

Our study also found endophthalmitis as an independent risk factor of stroke in AS patients (Table 4). It is unclear why endophthalmitis may be a risk factor for stroke development in AS patients. The characteristic of endophthalmitis is inflammation of the intraocular fluids and tissues. The immune response to endophthalmitis induces cell activation and cytokine secretion [24]. These cytokines include interferons, interleukins, tumor necrosis factors, and a number of growth factors, which are also related to stroke. Inflammation plays a significant role in stroke pathogenesis [25]. The immune system in the pathophysiology of stroke is complex. In the acute inflammatory phase, innate immune cells encroach upon the brain, leading to ischemic damage. Simultaneously, damaged brain cells release danger signals into the circulation that contribute to activation of the systemic immunity, followed by immunodepression [26]. In the chronic inflammatory phase, an adaptive immune response is initiated by antigen presentation targeting the brain, which would cause neuropsychiatric sequelae [26] because inflammation plays an important role in stroke and endophthalmitis may be related to cerebrovascular disease. Nonetheless, we are not fully aware of the mechanism that may contribute to a higher stroke risk with endophthalmitis in patients with AS, and more detailed studies must clarify this correlation.

Moreover, we divided the stroke patients into hemorrhagic and ischemic stroke groups (Table 5). In this study, the development of hemorrhagic or ischemic stroke was increased in AS patients with endophthalmitis. However, we did not compare the two groups using statistical analyses. The slightly increased aHR in the hemorrhagic stroke group should be interpreted carefully because of the lack of further statistical information.

Our study has several strengths. First, we compared the cumulative incidence of stroke between the study and comparison cohorts over a long-term study period using a longitudinal data analysis of the NHIRD system created in 1995. In addition, the coverage rate of the NHI in Taiwan was almost 99% because all citizens are obligated to enroll [27]. Hence, the data in our study were derived from a validated population-based database in Taiwan. A further strength of our study was its use of univariate and multivariate Cox regression analyses to adjust for confounding factors to ensure reliability of the results.

Nevertheless, there were some limitations in our study. First, it was a retrospective cohort study. Second, the NHIRD database lacks laboratory data, such as complete blood count with differential, erythrocyte sedimentation rate, or C-reactive protein, as advocatory evidence. It also lacks laboratory data, slit lamp findings, and imaging examinations, such as fundoscopy findings, which could be helpful for confirming the diagnosis of endophthalmitis and magnetic resonance imaging findings, which would also have revealed a correlation between AS and endophthalmitis. Third, all participants were Taiwanese; therefore, our findings may not be applicable to other ethnicities and countries. Fourth, there was a selection bias in our study because the cohort enrollment was limited to stroke patients. Finally, the claims database of our study is initially for health insurance statistics, not for research. Thus, our results might need to be validated by further research.

## 5. Conclusions

In conclusion, endophthalmitis could lead to an increased stroke risk in AS patients, among whom endophthalmitis was an independent risk factor for stroke. By reason of the risk of increasing potential cerebrovascular events, physicians should pay attention to AS patients with endophthalmitis. Further research is needed to clarify the mechanisms underlying endophthalmitis and stroke in patients with AS.

## Figures and Tables

**Figure 1 ijerph-19-13108-f001:**
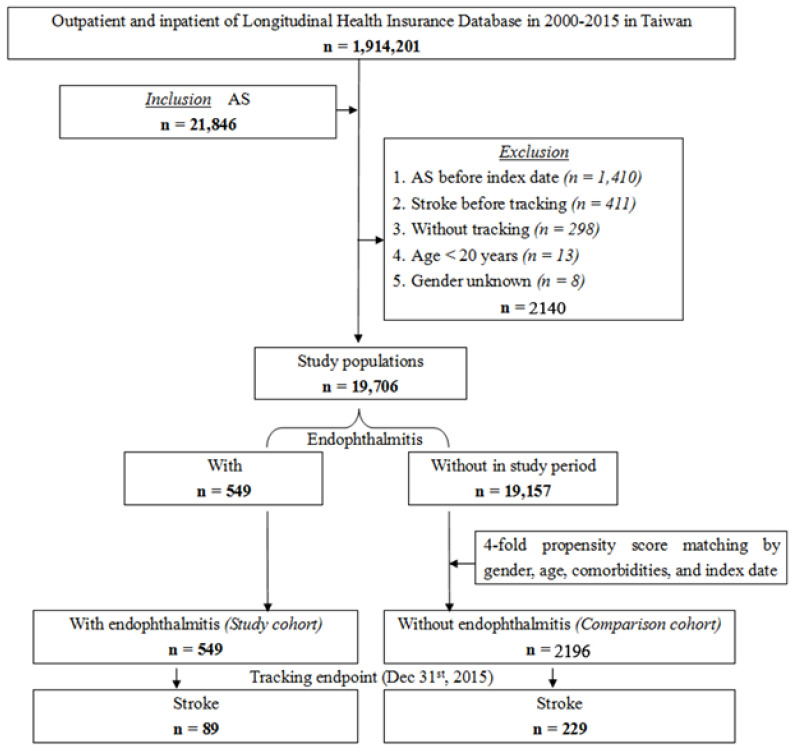
Algorithm of patient selection process. The inclusion criterion was ankylosing spondylitis. The exclusion criteria were diagnosis with ankylosing spondylitis before the inclusion date, stroke before tracking, lack of tracking, age < 20 years, and unknown sex.

**Figure 2 ijerph-19-13108-f002:**
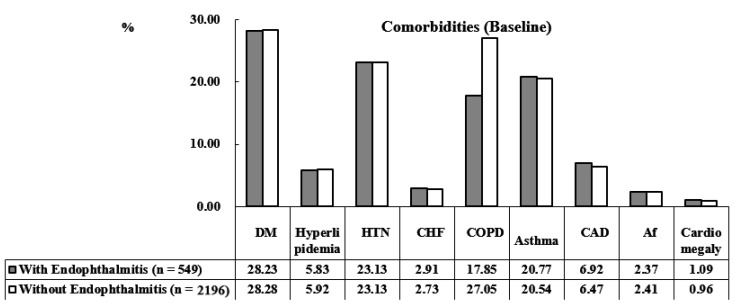
Bar graph of the comorbidities in the study population at baseline.

**Figure 3 ijerph-19-13108-f003:**
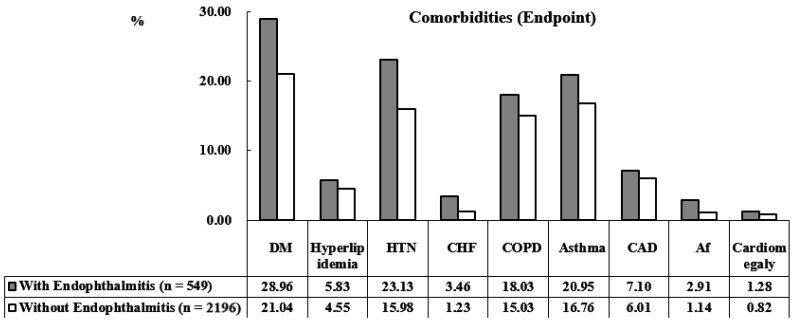
Bar graph of the comorbidities in the study population at study end.

**Figure 4 ijerph-19-13108-f004:**
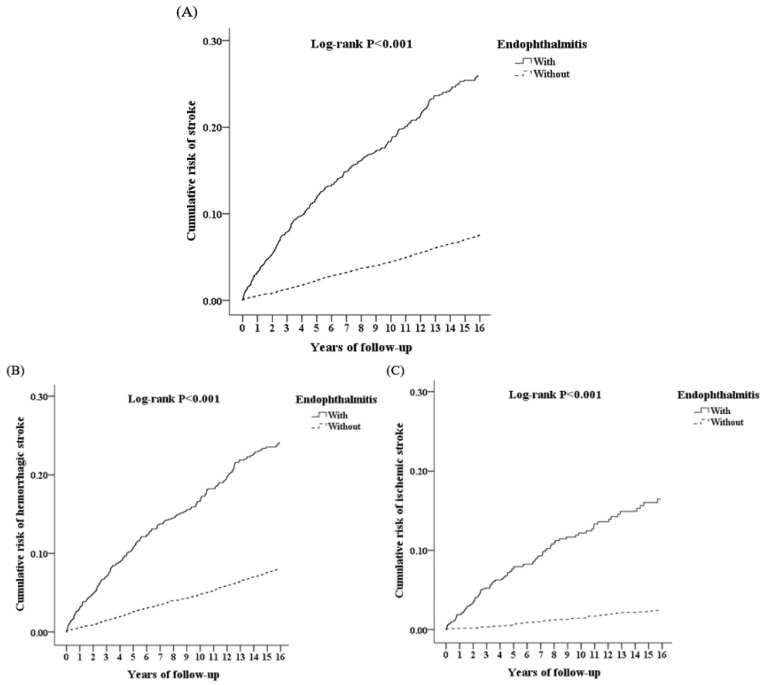
Kaplan–Meier analysis of the cumulative risk of stroke (**A**), hemorrhagic stroke (**B**), and ischemic stroke (**C**) among patients aged ≥ 20 years stratified by endophthalmitis status using the log-rank test.

**Figure 5 ijerph-19-13108-f005:**
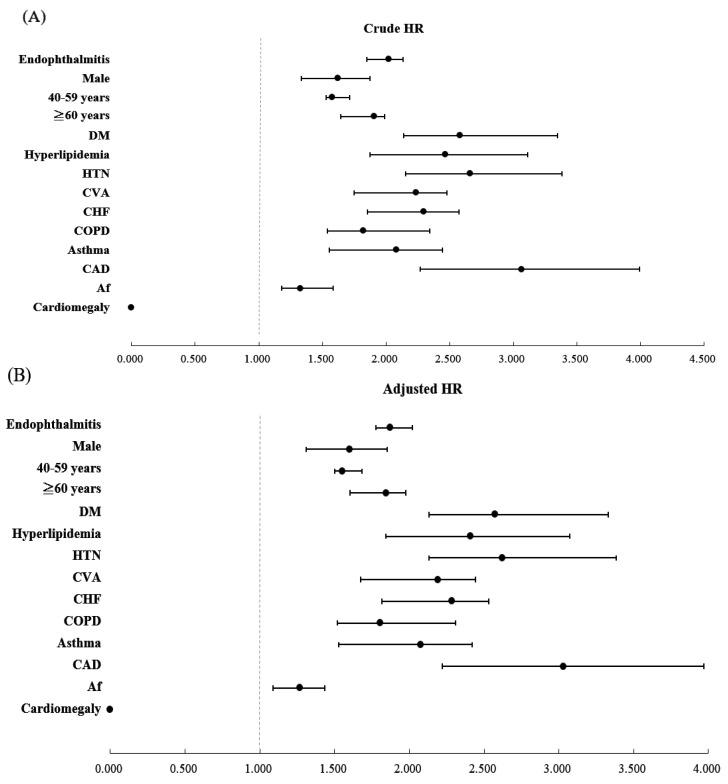
Forest plots of crude (**A**) and adjusted (**B**) hazard ratios for stroke factors evaluated in Table 3.

**Table 1 ijerph-19-13108-t001:** Baseline characteristics of the study population.

Endophthalmitis	Total		With		Without		*P*
Variables	*n*	%	*n*	%	*n*	%	
Total	2745		549	20.00	2196	80.00	
Gender							0.999
Male	1535	55.92	307	55.92	1228	55.92	
Female	1210	44.08	242	44.08	968	44.08	
Age (years)	37.07 ± 19.23		37.00 ± 18.65		37.08 ± 19.80		0.892
Age group (yrs)							0.999
20–39	1600	58.29	320	58.29	1280	58.29	
40–59	765	27.87	153	27.87	612	27.87	
≥60	380	13.84	76	13.84	304	13.84	
DM							0.922
Without	1969	71.73	394	71.77	1575	71.72	
With	776	28.27	155	28.23	621	28.28	
Hyperlipidemia							0.857
Without	2583	94.10	517	94.17	2066	94.08	
With	162	5.90	32	5.83	130	5.92	
HTN							0.935
Without	2110	76.87	422	76.87	1688	76.87	
With	635	23.13	127	23.13	508	23.13	
CHF							0.850
Without	2669	97.23	533	97.09	2136	97.27	
With	76	2.77	16	2.91	60	2.73	
COPD							0.883
Without	2053	74.79	451	82.15	1602	72.95	
With	692	25.21	98	17.85	594	27.05	
Asthma							0.752
Without	2180	79.42	435	79.23	1745	79.46	
With	565	20.58	114	20.77	451	20.54	
CAD							0.582
Without	2565	93.44	511	93.08	2054	93.53	
With	180	6.56	38	6.92	142	6.47	
Af							0.968
Without	2679	97.60	536	97.63	2143	97.59	
With	66	2.40	13	2.37	53	2.41	
Cardiomegaly							0.569
Without	2718	99.02	543	98.91	2175	99.04	
With	27	0.98	6	1.09	21	0.96	
CCI_R	0.91 ± 1.13		0.93 ± 1.19		0.90 ± 1.11		0.218

*P*: Chi-square/Fisher exact test on category variables and *t*-test on continuous variables. Af: atrial fibrillation, CAD: Coronary artery disease, CCI_R: Charlson comorbidity index revised, CHF: Congestive heart failure, COPD: Chronic obstructive pulmonary disease, DM: Diabetes Mellitus, HTN: Hypertension.

**Table 2 ijerph-19-13108-t002:** Patient characteristics at study end.

Endophthalmitis	Total		With		Without		*P*
Variables	*n*	%	*n*	%	*n*	%	
Total	2745		549	20.00	2196	80.00	
Stroke							<0.001
Without	2427	88.42	460	83.79	1967	89.57	
With	318	11.58	89	16.21	229	10.43	
Gender							0.999
Male	1535	55.92	307	55.92	1228	55.92	
Female	1210	44.08	242	44.08	968	44.08	
Age (yrs)	41.47 ± 19.97		41.24 ± 19.78		41.53 ± 20.02		0.613
Age group (yrs)							0.658
20–39	1556	56.70	310	56.45	1246	56.76	
40–59	756	27.53	148	27.04	607	27.65	
≥60	433	15.78	91	16.51	342	15.59	
DM							<0.001
Without	2124	77.38	390	71.04	1734	78.96	
With	621	22.62	159	28.96	462	21.04	
Hyperlipidemia							0.067
Without	2613	95.19	517	94.17	2096	95.45	
With	132	4.81	32	5.83	100	4.55	
HTN							<0.001
Without	2267	82.59	422	76.87	1845	84.02	
With	478	17.41	127	23.13	351	15.98	
CHF							<0.001
Without	2699	98.32	530	96.54	2169	98.77	
With	46	1.68	19	3.46	27	1.23	
COPD							<0.001
Without	2316	84.37	450	81.97	1866	84.97	
With	429	15.63	99	18.03	330	15.03	
Asthma							<0.001
Without	2262	82.40	434	79.05	1828	83.24	
With	483	17.60	115	20.95	368	16.76	
CAD							0.284
Without	2574	93.77	510	92.90	2064	93.99	
With	171	6.23	39	7.10	132	6.01	
Af							<0.001
Without	2704	98.51	533	97.09	2171	98.86	
With	41	1.49	16	2.91	25	1.14	
Cardiomegaly							0.240
Without	2720	99.09	542	98.72	2178	99.18	
With	25	0.91	7	1.28	18	0.82	
CCI_R	0.95 ± 1.13		0.98 ± 1.16		0.89 ± 1.11		0.026
All-caused mortality							0.002
Without	2461	89.65	475	86.52	1986	90.44	
With	284	10.35	74	13.48	210	9.56	

*P*: Chi-square/Fisher exact test on category variables and *t*-test on continuous variables. Af: atrial fibrillation, CAD: Coronary artery disease, CCI_R: Charlson comorbidity index revised, CHF: Congestive heart failure, COPD: Chronic obstructive pulmonary disease, DM: Diabetes Mellitus, HTN: Hypertension.

**Table 3 ijerph-19-13108-t003:** Factors of stroke by using Cox regression analysis.

Variables	Crude HR	95% CI	95% CI	*P*	Adjusted HR	95% CI	95% CI	*P*
Endophthalmitis								
Without	Reference				Reference			
With	2.021	1.850	2.135	<0.001	1.873	1.776	2.022	<0.001
Gender								
Male	1.621	1.333	1.874	<0.001	1.602	1.311	1.852	<0.001
Female	Reference				Reference			
Age (yrs)								
20–39	Reference				Reference			
40–59	1.576	1.530	1.713	<0.001	1.553	1.502	1.685	<0.001
≥60	1.906	1.644	1.991	<0.001	1.843	1.602	1.975	<0.001
DM								
Without	Reference				Reference			
With	2.580	2.141	3.348	<0.001	2.572	2.132	3.330	<0.001
Hyperlipidemia								
Without	Reference				Reference			
With	2.469	1.872	3.112	<0.001	2.410	1.841	3.072	<0.001
HTN								
Without	Reference				Reference			
With	2.662	2.156	3.385	<0.001	2.623	2.130	3.381	<0.001
CVA								
Without	Reference				Reference			
With	2.236	1.752	2.478	<0.001	2.191	1.672	2.440	<0.001
CHF								
Without	Reference				Reference			
With	2.297	1.855	2.573	<0.001	2.286	1.817	2.532	<0.001
COPD								
Without	Reference				Reference			
With	1.824	1.539	2.345	<0.001	1.803	1.519	2.311	<0.001
Asthma								
Without	Reference				Reference			
With	2.082	1.554	2.443	<0.001	2.076	1.529	2.419	<0.001
CAD								
Without	Reference				Reference			
With	3.068	2.267	3.991	<0.001	3.031	2.218	3.971	<0.001
Af								
Without	Reference				Reference			
With	1.326	1.178	1.586	<0.001	1.267	1.090	1.433	<0.001
Cardiomegaly								
Without	Reference				Reference			
With	0.000	-	-	0.999	0.000	-	-	0.999
CCI_R	1.649	1.602	1.660	<0.001	1.625	1.595	1.649	<0.001

Adjusted HR: Adjusted variables listed in the table. ACS: Acute coronary syndrome, Af: atrial fibrillation, CAD: Coronary artery disease, CCI_R: Charlson comorbidity index revised, CHF: Congestive heart failure, CI: confidence interval, COPD: Chronic obstructive pulmonary disease, CVA: Cerebrovascular accident, DM: Diabetes Mellitus, HR: hazard ratio, HTN: Hypertension, MetS: Metabolic syndrome.

**Table 4 ijerph-19-13108-t004:** Factors of stroke stratified by study variable using Cox regression analysis.

Endophthalmitis	With			Without (Reference)						
Stratified	Events	PYs	Rate (per 10^5^ PYs)	Events	PYs	Rate (per 10^5^ PYs)	Adjusted HR	95% CI	95% CI	*P*
Total	89	5170.50	1724.44	229	21,103.84	1085.11	1.873	1.776	2.022	<0.001
Gender										
Male	59	2895.28	2043.40	118	11,814.92	998.74	2.455	2.328	2.651	<0.001
Female	30	2275.22	1318.55	111	9288.92	1194.97	1.270	1.205	1.371	<0.001
Age (yrs)										
20–39	57	2918.36	1953.15	153	11,987.22	1276.36	1.787	1.695	1.929	<0.001
40–59	26	1390.02	1870.48	65	5838.24	1113.35	1.900	1.802	2.051	<0.001
≥60	6	862.12	714.76	11	3278.38	335.53	3.284	3.114	3.546	<0.001
DM										
Without	55	3705.28	1488.74	184	16,640.57	1105.73	1.571	1.490	1.695	<0.001
With	34	1465.22	2320.47	45	4463.27	1008.23	2.791	2.647	3.014	<0.001
Hyperlipidemia										
Without	62	4867.61	1277.06	190	20,155.82	942.66	1.571	1.490	1.695	<0.001
With	27	302.89	8914.13	39	948.02	4113.84	2.734	2.592	2.952	<0.001
HTN										
Without	53	3973.26	1338.00	183	17,668.73	1035.73	1.522	1.444	1.644	<0.001
With	36	1197.24	3006.92	46	3435.11	1339.11	2.632	2.496	2.841	<0.001
CVA										
Without	60	4872.48	1234.73	193	20,013.39	964.35	1.502	1.424	1.621	<0.001
With	29	298.02	9730.89	36	1090.45	3301.39	3.591	3.404	3.877	<0.001
CHF										
Without	77	5031.25	1533.66	214	20,831.98	1027.27	1.761	1.669	1.900	<0.001
With	12	139.25	8617.59	15	271.86	5517.55	1.897	1.799	2.048	<0.001
COPD										
Without	78	4242.07	1842.55	211	17,901.39	1178.68	1.831	1.736	1.977	<0.001
With	11	928.43	1184.80	18	3202.45	562.07	2.774	2.631	2.994	<0.001
Asthma										
Without	72	4104.26	1758.22	208	17,635.59	1179.43	1.761	1.670	1.901	<0.001
With	17	1066.24	1594.39	21	3468.25	605.49	3.050	2.893	3.293	<0.001
CAD										
Without	79	4820.06	1642.35	219	19,814.43	1105.26	1.761	1.670	1.901	<0.001
With	10	350.44	2853.56	10	1289.41	775.55	3.952	3.747	4.266	<0.001
Af										
Without	86	5028.24	1713.56	227	20,975.81	1082.20	1.865	1.767	2.012	<0.001
With	3	142.26	2108.81	2	128.03	1562.13	2.367	2.245	2.556	<0.001
Cardiomegaly										
Without	89	5100.24	1748.19	229	21,009.16	1090.00	1.873	1.776	2.022	<0.001
With	0	70.26	0.00	0	94.68	0.00	-	-	-	-

PYs = Person-years; Adjusted HR = Adjusted Hazard ratio: Adjusted for the variables listed in Table 3; CI = confidence interval.

**Table 5 ijerph-19-13108-t005:** Factors of stroke subgroup by using Cox regression analysis.

Endophthalmitis	With			Without (Reference)						
Stroke Subgroup	Events	PYs	Rate (per 10^5^ PYs)	Events	PYs	Rate (per 10^5^ PYs)	Adjusted HR	95% CI	95% CI	*P*
Overall	89	5170.5	1724.44	229	21,103.84	1085.11	1.873	1.776	2.022	<0.001
Hemorrhagic stroke	47	5170.5	909.00	119	21,103.84	563.88	1.998	1.895	2.153	<0.001
Ischemic stroke	42	5170.5	812.30	110	21,103.84	521.23	1.794	1.692	1.924	<0.001

PYs = Person-years; Adjusted HR = Adjusted Hazard ratio: Adjusted for the variables listed in Table 3.; CI = confidence interval.

## Data Availability

Data available in a publicly accessible repository. The data presented in this study are openly available in FigShare at https://figshare.com/s/e6ae1d6e57885b01d7c2 (accessed on 14 August 2022).

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
