# Peer review of "Correlation between Endophthalmitis and Stroke Development in Ankylosing Spondylitis Patients: A Population-Based Cohort Study"

_ijerph, 2022, doi:10.3390/ijerph192013108_

Round 1

Reviewer 1 Report

The article: ”Correlation between Endophthalmitis and the Stroke Development in Ankylosing Spondylitis Patients: A Population-Based Cohort Study” aimed to research the correlation between endophthalmitis and stroke development in ankylosing spondylitis (AS) patients. It was based on data: National Health Insurance Research Database (NHIRD). The relationship between the occurrence of strokes and ankylosing spondylitis has been well documented in recent years. A novel approach in this study is the inclusion of endophthalmitis as an independent risk factor for strokes in AS patients. Nobody has studied the relationship between these diseases before (except for a case report - source [9]). The study was very well designed, taking into account good design practices for this type of study, e.g. the control group 4 times larger than the study group. The research methods were also well selected: statistical tests, which allowed for obtaining reliable results. Possible weaknesses of the article were disclosed by the authors at the end of the "Discussion". The topic was very well thought out, presented and discussed using references to previous research. There has also been an attempt to biochemically explain the correlation: endophthalmitis versus stroke. The conclusions were correctly formulated.

I would just make a few small technical adjustments:

- line 42 - delete: "[1-4]", because the next sentence also contains the same reference.

- Figure: 4,5 - larger figures, too small font used, which makes the figures difficult to read. All figures should be of similar font size.

- In the chapter: "2.3 Ethical considerations" - sentences: "Our study was approved by the Institutional Review Board of the Tri-Service General Hospital (TSGHIRB no. E202216009). The study protocol met the ethical guidelines of the 1975 Declaration of Helsinki ”. are unnecessary because the relevant information is included at the end of this article: Institutional Review Board Statement.

- If possible, the cited literature could be refreshed.

Author Response

Dear Reviewer:

Thank you for allowing us to submit a revised draft of our manuscript entitled,” Correlation between Endophthalmitis and the Stroke Development in Ankylosing Spondylitis Patients: A Population-Based Cohort Study.” to International Journal of Environmental Research and Public Health. The previous manuscript ID is: ijerph- 1891602. We also appreciate the time and effort you and each of the Reviewers have dedicated to providing constructive feedback.

Our responses to the Reviewers’ comments and suggestions are described below in a point-to-point manner. Appropriate changes, suggested by the Reviewers, has been introduced to the manuscript.

Point 1: line 42 - delete: "[1-4]", because the next sentence also contains the same reference.

Response 1: We appreciate your suggestion. We deleted "[1-4]" in line 42.

Point 2: Figure: 4,5 - larger figures, too small font used, which makes the figures difficult to read. All figures should be of similar font size.

Response 2: We appreciate your suggestion. We revised the Figure 4 and Figure 5.

Point 3: In the chapter: "2.3 Ethical considerations" - sentences: "Our study was approved by the Institutional Review Board of the Tri-Service General Hospital (TSGHIRB no. E202216009). The study protocol met the ethical guidelines of the 1975 Declaration of Helsinki ”. are unnecessary because the relevant information is included at the end of this article: Institutional Review Board Statement.

Response 3: We appreciate your suggestion. We deleted "Our study was approved by the Institutional Review Board of the Tri-Service General Hospital (TSGHIRB no. E202216009). The study protocol met the ethical guidelines of the 1975 Declaration of Helsinki ” in the chapter: "2.3 Ethical considerations".

Point 4: If possible, the cited literature could be refreshed.

Response 4: We appreciate your suggestion. We checked all the cited literature again and they were correct.

Thanks a lot again and your comments and suggestions really helped us to improve our manuscript better. We hope that our manuscript will be acceptable for publication in International Journal of Environmental Research and Public Health.

We look forward to hearing from you.

Sincerely,

Ching-Long Chen

Department of Ophthalmology, Tri-Service General Hospital, National Defense Medical Center, Taipei, Taiwan.

No. 325, Sec. 2, Cheng Gong Rd, Nei-Hu District, Taipei 114, Taiwan.

TEL: +886-2-8792-7164 

FAX: +886-2-8792-7104

Reviewer 2 Report

I do not think correlation between endophthalmitis and the stroke in AS patients can be established by analysis of data from this claims database. Numerous factors including the severity of AS or other comorbidities, which could not be investigated in their study, strongly influence the occurrence of stroke (maybe rather than endophtalmitis).

Considering the presence of such unknown confounders and an issue from time-varying covariates, their statistical method, Cox regression with Fine and Gray’s competing risk model is not appropriate for this longitudinal study with long term follow-up.

In addition, claims database is not originally designed for research, and have several limitations for clinical research. However, they did not show any efforts to validate their results.

Author Response

Dear Reviewer:

Thank you for your careful review of our manuscript entitled,” Correlation between Endophthalmitis and the Stroke Development in Ankylosing Spondylitis Patients: A Population-Based Cohort Study.” The previous manuscript ID is: ijerph- 1891602. We also appreciate the time and effort you and each of the Reviewers have dedicated to providing constructive feedback.

Our responses to the Reviewers’ comments are described below in a point-to-point manner. Appropriate changes, suggested by the Reviewers, has been introduced to the manuscript.

Point 1: I do not think correlation between endophthalmitis and the stroke in AS patients can be established by analysis of data from this claims database. Numerous factors including the severity of AS or other comorbidities, which could not be investigated in their study, strongly influence the occurrence of stroke (maybe rather than endophtalmitis).

Response 1: We appreciate your constructive comments. In our study, AS patients with asthma, Af, COPD, CHF, CAD, DM, hyperlipidemia, HTN, male, aged 40–59 years and ≥60 years were at higher risk of developing stroke which presented in Table 3. After stratified analysis comparing AS patients with and without endophthalmitis, we found that above factors were still at higher risk of developing stroke which presented in Table 4. Our study also found endophthalmitis as an independent risk factor of stroke in AS patients. However, it is unclear why endophthalmitis may be a risk factor for developing stroke in AS patients. Therefore, the detailed mechanism needs further research to identify in the future.

Point 2: Considering the presence of such unknown confounders and an issue from time-varying covariates, their statistical method, Cox regression with Fine and Gray’s competing risk model is not appropriate for this longitudinal study with long term follow-up.

Response 2: We appreciate your constructive comments. The Cox proportional-hazards model is essentially a regression model commonly used statistical in medical research for investigating the association between the survival time of patients and one or more predictor variables.[1] According to your comments, we deleted Fine and Gray’s competing risk model and revised Table 3, 4 and 5 and associated data in our manuscript.

Reference 1:

David R Cox. 1972. Regression models and life-tables. Journal of the Royal Statistical Society: Series B (Methodological) 34, 2 (1972), 187–202. https://doi.org/10.1111/j.2517-6161.1972.tb00899.x

Point 3: In addition, claims database is not originally designed for research, and have several limitations for clinical research. However, they did not show any efforts to validate their results.

Response 3: We appreciate your constructive comments. According to your comment, we added a limitation in Line 229-230: “The claims database of our study is initially for health insurance statistics, not for research. Thus, our results might need to be validated by further research.”

Thanks a lot again and your comments and suggestions really helped us to improve our manuscript better. We hope that our manuscript will be acceptable for publication in International Journal of Environmental Research and Public Health.

We look forward to hearing from you.

Sincerely,

Ching-Long Chen

Department of Ophthalmology, Tri-Service General Hospital, National Defense Medical Center, Taipei, Taiwan.

No. 325, Sec. 2, Cheng Gong Rd, Nei-Hu District, Taipei 114, Taiwan.

TEL: +886-2-8792-7164 

FAX: +886-2-8792-7104
